# Exploring Biomarkers of Mental Flexibility in Healthy Aging: A Computational Psychometric Study

**DOI:** 10.3390/s23156983

**Published:** 2023-08-06

**Authors:** Francesca Borghesi, Alice Chirico, Elisa Pedroli, Giuseppina Elena Cipriani, Nicola Canessa, Martina Amanzio, Pietro Cipresso

**Affiliations:** 1Department of Psychology, University of Turin, Via Verdi 10, 10124 Turin, Italy; giuseppinaelena.cipriani@unito.it (G.E.C.); martina.amanzio@unito.it (M.A.); pietro.cipresso@unito.it (P.C.); 2Department of Psychology, Research Center in Communication Psychology, Universitá Cattolica del Sacro Cuore, 20123 Milan, Italy; alice.chirico@unicatt.it; 3Faculty of Psychology, eCampus University, 22060 Novedrate, Italy; elisa.pedroli@uniecampus.it; 4Department of Geriatrics and Cardiovascular Medicine, IRCCS Istituto Auxologico Italiano, 20149 Milan, Italy; 5ICoN Center, Scuola Universitaria Superiore IUSS, 27100 Pavia, Italy; nicola.canessa@iusspavia.it; 6Istituti Clinici Scientifici Maugeri IRCCS, Cognitive Neuroscience Laboratory of Pavia Institute, 27100 Pavia, Italy; 7Istituto Auxologico Italiano, IRCCS, 20145 Milan, Italy

**Keywords:** mental flexibility, psychometric model, affect dynamics, bio-markers, neuroscience

## Abstract

Mental flexibility (MF) has long been defined as cognitive flexibility. Specifically, it has been mainly studied within the executive functions domain. However, there has recently been increased attention towards its affective and physiological aspects. As a result, MF has been described as an ecological and cross-subject skill consisting of responding variably and flexibly to environmental cognitive-affective demands. Cross-sectional studies have mainly focused on samples composed of healthy individual and of patients with chronic conditions such as Mild Cognitive Impairment and Parkinson’s, emphasizing their behavioral rigidity. Our study is the first to consider a sample of healthy older subjects and to outline physiological and psychological markers typical of mental flexibility, to identify functional biomarkers associated with successful aging. Our results reveal that biomarkers (respiratory and heart rate variability assessments) distinguished between individuals high vs. low in mental flexibility more reliably than traditional neuropsychological tests. This unveiled the multifaceted nature of mental flexibility composed of both cognitive and affective aspects, which emerged only if non-linear multi-variate analytic approaches, such as Supervised Machine Learning, were used.

## 1. Introduction

Aging is an inevitable and complex process that brings about a myriad of changes in both cognitive and emotional domains. While some individuals experience cognitive decline and emotional instability as they age, a growing body of evidence suggests that many older adults maintain remarkable cognitive and emotional health levels, often called *successful aging* [1]. Understanding the factors that contribute to successful aging has become a focal point in gerontological research, as it not only enhances our knowledge of aging but also informs interventions to promote healthy aging outcomes.

In recent years, researchers have focused on investigating the role of cognitive and affective flexibility in successful aging [1,2]. Theoretically, these studies introduced flexibility as a unitary concept composed of both cognitive and affective components. However empirically, these studies did not test these components together in a sample of healthy elderly individuals. In fact, the only studies that analyzed and defined these components have focused on young, healthy subjects or individuals with already established neurological conditions such as Mild Cognitive Impairment, Parkinson’s, or Alzheimer’s disease [3,4,5,6,7,8,9,10,11]. Conversely, this study aims to experimentally investigate the biomarkers of cognitive-affective flexibility in a sample of healthy elderly individuals. This exploratory study seeks to identify which bio-psyco components distinguish between healthy aging individuals with high vs. low levels of flexibility, to identify functional biomarkers associated with successful aging.

First, Mental Flexibility is a concept everyone is familiar with but is difficult to define clearly and unambiguously. Similarly, also at the scientific level, a partial definition of its cognitive and affective components is usually provided rather than a general and comprehensive definition [4,5,12,13,14,15,16,17]. The following are usually identified among MF’s components: Cognitive Flexibility, Affective Flexibility, and its application in young and patient older adult sampling.

Cognitive Flexibility (CF) is the ability to adjust goals and shift a course of thought or action according to the changing demands of the situation. It involves two central components of executive control: inhibition and set-shifting [4,5,15,18,19]. Inhibition is the ability to override prepotent responses and inhibit processing irrelevant material and goals. Set shifting is the ability to flexibly shift focus between goals or mental sets [4,20,21]. Affective Flexibility (AF) is defined as a subset of cognitive flexibility and it is expressed in terms of speed and correctness in discriminating between emotional stimuli [13,14]. In fact, it involves the specific ability to switch between emotion-focused and non-emotional cognitive sets. For example, reappraising an emotion-eliciting situation often requires the ability to change perspective and put the situation into a broader or different context to change the perceived importance of the situation. This involves focusing on emotionally neutral information within and outside of the situation [13,14,22].

CF is conventionally measured by neuropsychological tests (e.g., Wisconsin Card Sorting Test or Trail Making Test) and self-report questionnaires (e.g., Cognitive Flexibility Inventory or Cognitive Flexibility Scale) [5,23]. Neuropsychological tests assess CF in terms of attention, visual screening ability, and processing speed. In contrast, self-reports provide an ecological assessment of relational and communicative skills [5,19,24]. Several recent studies have shown that, while they theoretically measured the same construct, the measures did not correlate significantly [5,17,20,24,25]. Most studies stem from a neuropsychology view and interpret AF as a cognitive flexibility property [13,14,22]. AF is studied in terms of the task-switching paradigm. The affective task-switching paradigm required participants to shift between categorizing positive and negative affective pictures according to emotional or non-emotional features [13,14,22]. Previous research showed that greater affective flexibility (less switch costs) predicted the ability to use reappraisal to down-regulate emotions but was linked to a lower resting state Heart Rate Variability (HRV) [14]. This result is controversial since, usually, higher resting of heart rate variability corresponds to high quality of life and high emotional regulation abilities [14,26,27].

Previous studies first sought to explain considered CF only for executive functions, AF as a subcomponent, and considering only sampling of young adults or patients with chronic disease [4,5,21,25]. Those studies addressing CF measured shifting abilities in a clinical setting, poorly generalizable to flexible daily living skills, analyzed instead by self-reports. Furthermore, when AF is viewed as a subcomponent of CF, it loses some of its emotional and variable significance. This way, AF is discretized and made stimulus-responsive, not illustrative of emotional variability. Finally, its various definitions and measurements have mainly been studied with young samples or chronic disease.

Hence, our study is the first to consider MF in older healthy aging subjects, trying to outline its psycho-physiological components in a successful aging sample. At the cognitive and affective level, flexibility is the ability to act and change behavior variably, adapting to the environment. This variability has been measured cognitively by neuropsychological tests and self-reports, and affectively measured by physiological signals during emotional imagery administration. The idea is to measure Affective Flexibility, expressed through continuous affective dynamics [28,29,30,31,32,33]. Put differently, here, we measured the cognitive side of MF using conventional neuropsychological tests, and the affective dimension using psychophysiological signals.

Considering flexibility in terms of cognitive-affective and physiological variability in a healthy aging population could allow us outlining the most accurate psycho-bio markers of flexibility and how they interact with other measurements of aging.

## 2. Materials and Methods

### 2.1. Participants

A group of 35 (27 women; 62–82 years) socially active older people from the University of the Third Age (UNITRE) in Turin (Italy) were recruited, and all agreed to participate. A priori power analysis was conducted using G*Power version 3.1.9.7 [34] to determine the minimum sample size required to test the study hypothesis, based on Fernandez-Aguilar et al. [35]. Results indicated the required sample size to achieve 95% power for detecting a large effect, at a significance criterion of α = 0.05, was N = 26 for independent samples t-test (High Flex vs. Low Flex). Considering dropouts and recording issues with physiological signals, we recruited 35 participants. All 35 subjects were >60 years old and, thus, can be classified as “older adults” (World Health Organization. Health Topics, Ageing. https://www.who.int/health-topics/ageing#tab=tab1, accessed on 1 July 2023). The study was conducted in accordance with the Declaration of Helsinki, having been approved by the Ethics Committee of the University of Turin (Prot. No. 10038 and 151786).

### 2.2. Inclusion Criteria

All participants gave written informed consent before participating in the study. In particular, to be included in the study, participant criteria were: >60 years old in order to be classified as “older adults” (World Health Organization); no neurological or psychiatric disorders and, consequently, were not taking any psychotropic medications that could have affected their cognitive abilities or mood (based on the Cumulative Illness Rating Scale (CIRS) [36]). None of the participants met Fried et al.’s inclusion criteria for frailty status [37], based on five parameters, weight loss, self-reported fatigue, decreased physical activity, grip strength, and walking speed. The absence of all criteria characterized robust subjects, the presence of one or two described a prefrail status, and three or more represented a frail individual. Most participants were classified as “robust” and “prefrail” (65.71% and 34.29%, respectively) [37].

### 2.3. Procedure

Participants in the study were enrolled in UniTRE educational modules, through convenience sampling. They were healthy elderly students who paid an annual membership fee. The main purpose of UniTRe regards Informal programs such as the Universities of the Third Age to promote lifelong learning and pursuing personal interests and goals, a key element of healthy aging. Of 200 students, approximately 17% of the total sample, i.e., 35 socially active subjects who did not complain of subjective cognitive decline, were voluntarily recruited based on the inclusion criteria. The participants who met the experimental criteria for healthy aging were contacted through email and/or telephone to plan a meeting. Participants were requested to sit in front of a computer, and they were taught about the basic aims of the research and the techniques to be employed. It is a two-step study: a first phase of administration of a Neuropsychological battery, to test Cognitive Flexibility, and a second phase of psychophysiological test, to test Affective Flexibility through administering emotional stimuli, depicting the initial and most stressful phase of the COVID-19 pandemic.

#### 2.3.1. First Phase: Assessing Cognitive Flexibility through Neuropsychological Tests

Participants completed neuropsychological battery in 90 min. To avoid fatigue, the assessment was divided into two parts, lasting approximately 45 min, and held on two different days. The neuropsychological battery was composed of specific measures for CF and concurrent measurements, considered key for the definition of CF (Table 1), validated in Italian young and older healthy sampling [7,16,18]. We assessed CF with neuropsychological tests, as executive function, and with self-report to test CF in everyday life settings.

For neuropsychological tests we used the subscales of Memory (ACE-M), Visual spatial (ACE-VS), Attention/Orientation (ACE-AO) of Addenbrooke’s Cognitive Examination (ACE) [38], Montreal Cognitive Assessment (MOCA) [39] and Trail Making Task (TMT) [40], as indicated in the literature [19].

The Addenbrooke’s Cognitive Examination (ACE) is a comprehensive neuropsychological test that evaluates various cognitive domains, including visuospatial abilities and memory functions. These cognitive domains are closely linked to cognitive flexibility through their impact on problem-solving and adaptive behaviors. The visuospatial and attention component of the ACE assess an individual’s capacity to perceive and process visual information, understand spatial relationships, and mentally manipulate objects in space. The memory component of the ACE evaluates an individual’s short-term and long-term memory abilities. Memory plays a crucial role in cognitive flexibility, allowing individuals to draw on past experiences, information, and learn strategies to adapt to new challenges. For example, recalling previous solutions or experiences and applying them in novel situations is essential for flexible problem-solving. These abilities are interconnected with cognitive flexibility, as they underpin an individual’s capability to rotate objects mentally between different visual perspectives and adapt their spatial strategies to changing situations [41].

The Montreal Cognitive Assessment (MOCA) is a widely used neuropsychological test designed to assess various cognitive domains, including cognitive flexibility. It is commonly employed as a screening tool to detect mild cognitive impairment and early signs of dementia. Within the MOCA, cognitive flexibility is evaluated through specific tasks that assess an individual’s ability to shift attention, mental set, and problem-solving strategies. For instance, the test may include tasks that require participants to switch between different cognitive rules or to alternate between different categories when performing verbal fluency exercises. Furthermore, the MOCA assesses working memory, another cognitive function closely related to cognitive flexibility. Working memory is essential for holding and manipulating information in mind, allowing individuals to adapt to new information and switch between tasks effectively [42].

The Trail Making Test (TMT) is a widely used neuropsychological assessment consisting of Part A and Part B. The test evaluates visual attention, processing speed, and cognitive shifting abilities as cognitive flexibility components. Part A primarily assesses visual attention and processing speed, as it requires tracking and quickly connecting the numbers visually. Part B is a more complex task, where the individual is required to alternate between connecting numbers and letters in ascending order while switching between the two sets (1, A, 2, B, etc.). Both sections specifically target cognitive flexibility, as it demands the ability to shift attention, switch between mental sets, and adapt between two different sequences, as the participants must rapidly and accurately switch between numbers and letters while maintaining the correct sequence. The task challenges the individual to inhibit a prepotent response (continuing with numbers) and flexibly shift to a new set of stimuli (switching to letters) [21].

As self-report measuring of CF, we used the validated Cognitive Flexibility Inventory (CFI) [43], measuring ecologically the ability to identify alternative solutions, generate several explanations, and perceive difficult conditions as controllable. The self-report inventory includes 19 items that assess an individual’s capacity to switch between different tasks, perspectives, or problem-solving approaches. Respondents are asked to rate their agreement or disagreement with each statement, providing insights into their cognitive adaptability and mental agility. The Cognitive Flexibility Inventory has been widely utilized in both research and clinical settings to understand the cognitive processes underlying adaptive behaviors and problem-solving skills [43,44,45]. Researchers often employ the CFI to investigate the role of cognitive flexibility in various domains, such as academic performance, emotional regulation, and decision-making [18,46,47,48,49,50,51].

Convergence measures of CF assessed different facets of cognitive and affect processing, already demonstrated to be significantly positively and inversely correlated with the construct of CF [4,5,52,53]. For affect processing Apathy Evaluation Scale (AES) [54], Beck Depression Inventory (BDI) [55], Hamilton Anxiety Rating Scale (HARS) [56], and Emotion Regulation Questionnaire (ERQ) [57] were administered. For cognitive processing, a Cumulative Illness Rating Scale (CIRS) [36] was used to assess medical history in terms of comorbidity and severity indexes (see Table 1), and a Cognitive Function Instrument (CF-Instrument) was used to assess subjective cognitive function measured by the subject and the partner [58].

#### 2.3.2. Second Phase: Assessing Affective Flexibility through Psychophysiological Signals

After being administered the neuropsychological assessments, participants were exposed to a set of selected emotional images depicting events related to the lockdown COVID-19 pandemic, to assess individual differences in Affective Flexibility [59]. We randomly administered 75 pictures featuring emotional content associated with the pandemic situation [59]. COVID-19-related images were selected to enhance their perceived personal relevance, thus potentially triggering a wider range of emotional responses, since the study was conducted between the first and second Italian lockdowns (late 2020 to early 2021). To explore the variations in physiological arousal associated with the COVID-19 pandemic, we conducted a search on the “Google images” website (https://images.google.com/ (accessed on 1 July 2023)) for pictures depicting the crucial period between the virus outbreak and the lockdown in Italy. We used the keyword “COVID” and initially obtained 124 images, which were then individually assessed by all authors to select those meeting specific inclusion criteria. The images should (i) include people of diverse ages, genders, and sociodemographic backgrounds; (ii) feature elements pertinent to the pandemic context, such as masks, gloves, and medical personnel; and (iii) represent typical COVID-19 everyday situations and hospital scenes. Then, (iv) images should depict at least two people interacting (“social” images). (v) To enhance familiarity and emotional relevance, the images should feature individuals with exclusively Caucasian facial features, matching the ethnicity of the study participants (“ethnicity”). Finally, to avoid duplicates, only unique images with distinct content, formats, and resolutions were included. This assessment lasted 45–60 min.

During this exposure, their psychophysiological activity was recorded, as a measure of Affective Flexibility. The idea is that AF refers to an individual’s ability to adapt and regulate emotional responses in response to changing situations or emotional stimuli, closely related to physiological variability, which refers to the variation in physiological responses, such as heart rate, skin conductance, and facial expressions, in different emotional contexts. Individuals with higher affective flexibility tend to exhibit greater physiological variability, as they can effectively adjust their physiological responses based on the emotional demands of the situation [60].

First, the researcher applied the sensor electrodes for Blood Volume Pulse (BVP) and facial Electromyography (f-EMG) (zygomatic and supercilii corrugator activity were measured). Finally, Skin Conductance (SC) was recorded with two electrodes applied to annular and index fingers. All participants were right-handed (without a history of switching the dominant hand during their lifetimes).

Then, a two-minute baseline session was conducted with all participants to establish a stable reference. The psychophysiological assessment began according with a specific trigger synchronized with experimental stimuli.

Finally, the experimenter helped the participants remove all the electrodes and patches, following a debriefing phase.

Using images to assess affective flexibility was already utilized in the literature [13,14,22]. However, in our experiment, the variability associated with affective flexibility is measured in terms of psychophysiological reactions to different emotional stimuli, rather than in terms of speed and accuracy in discriminating between different emotional stimuli as in previous studies.

### 2.4. Recording of Psychophysiological Signals

The data on the autonomic nervous systems were collected by measuring physiological responses, i.e., Blood Volume Pulse (BVP), Respiration (RSP), Facial Electromyography (Zygomatic and Corrugator) (fEMG), and Galvanic Skin Response (GSR). Nexus-4 acquired these responses. The responses were then processed with custom software developed using MATLAB 9.13.0 (R2022b) (The Mathworks, Inc., Natick, MA, USA). Every channel was acquired synchronously at 2048 Hz and extracted at 256 Hz for the computation of indices.

### 2.5. Psychophysiological Signal Processing

Cardiovascular and respiratory activities were monitored to assess the voluntary and autonomic effects of breathing on heart rate. We examined the Inter-Beat Interval (IBI) from the Blood Volume Pulse sensor, which is similar to the RR peaks interval from the ECG. Inter-beat interval (IBI, following also RR) was converted into an estimate of heart rate (HR) and pulse amplitude (BVP Amplitude), which indicate the proportionate increase in blood volume. BVP heart rate readings were represented as HR mean (beats per minute) and RR mean (60,000/HR). To assess autonomic nervous system response, the Task Force of the European Society of Cardiology and the North American Society of Pacing and Electrophysiology recommends extracting typical temporal, spectral, and non-linear Heart Rate Variability (HRV) indices [61]. Time-domain indices of HRV quantify the amount of variability in measurements of the inter-beat interval (IBI), which is the period between successive heartbeats. As a temporal domain measure, we calculated standard deviation of NN intervals (SDNN), minimum and maximum HR computed using N beat moving average (Max and Min HR), percentage of successive RR intervals that differ by more than 50 ms (pNN50), and the root mean square of successive RR interval differences (RMSSD) using the BVP IBI. For the frequency domain, spectral analysis was performed using Fourier spectral methods. In particular, Standard Heart Rate Variability (HRV) spectral-method indexes and similar indexes were used to evaluate the response of the autonomic nervous system. We calculated the magnitude of the peak frequency (also indicated as RR peak frequency) in the power spectrum. The rhythms were classified as very low frequency (VLF < 0.04 Hz), low-frequency (LF, between 0.04 and 0.15 Hz), and high frequency (HF, from 0.15 to 0.5 Hz) oscillations. This procedure also allowed us to calculate the LF/HF ratio, a well-known sympathovagal balance index. Nonlinear domain allows quantification of the unpredictability of a time series, plotting every R–R in a Poin-Carrè Plot. Poincaré plot analysis lets researchers visually explore hidden patterns in time series (a sequence of values from successive measurements). Poincaré plot analysis is insensitive to R–R interval trends, unlike frequency domain observations. We considered that the standard deviation (hence SD) of the distance of each point from the y = x axis (SD1) specifies the ellipse’s width, and the standard deviation of each point from the y = x + average R–R interval (SD2) specifies the ellipse’s length the ratio of SD1/SD2, which measures the unpredictability of the RR time series, used to measure autonomic balance [62,63].

The respiration signal was filtered to produce a smooth sinusoidal signal [64]. The Respiration Period index represents the peak-to-peak time (maximum-to-maximum distance of the sinusoid), which allowed us to compute the Respiration Rate (RSP Rate) that corresponded to the breaths per minute.

Skin Conductance (SC) or Skin Conductance Response (SCR) can be extracted from a Galvanic Skin Response (GSR) biosensor. Electrodermal activity is measured in conductance (microsiemens). SCR may be captured at 32 Hz without distortion because it is a slow physiological function. The mean of the sampled signal after artifact reduction was used to construct SC’s mean index [65].

The raw electromyography (EMG raw) is a collection of positive and negative electrical signals; their frequency and amplitude give us information on the contraction or rest state of the muscle. The Root Mean Square (RMS) is generally considered for rectifying the raw signal and converting it to an amplitude envelope [66]. We considered both corrugator electromyography (EMG1) and zygomatic electromyography (EMG2).

### 2.6. Statistical Analyses

Analyses were performed using Jamovi Statistics software (version 2.2.5.0). Two normality tests (i.e., Kolmogorov–Smirnov and Shapiro–Wilk) were performed to determine whether the variables were normally distributed. Conditions (Low Flexibility-High Flexibility) were compared using independent *t*-tests.

### 2.7. Computational Analyses

Computational analyses were carried out using Python 3.4 with the Orange 3.34 data mining suite, which was available free in the open-source code (https://github.com/biolab/orange3 (accessed on 1 July 2023) and from which it is possible to see all the algorithms used in the article. In particular, cross-validation leave one out was performed using the following methods [44,45], i.e., (1) Random Forest classification using an ensemble of decision trees; (2) Support Vector Machine (SVM) to map inputs to higher-dimensional feature spaces that best separate different classes; and (3) Naïve Bayes, a probabilistic classifier based on Bayes’ theorem and K-Nearest Neighbors (kNN) to predict according to the nearest training instances, with Euclidean metric and uniform weight. As stated before, all the algorithms used were available in the open-source code and documentation related to them can be found in the Scikit user guide, which provides a detailed explanation of all the algorithms used in the study, including rank calculation, classification tree, and learners (http://scikit-learn.org/stable/user_guide.html (accessed on 1 July 2023).

## 3. Results

In the first analysis, we used classical null hypothesis significance testing (NHST) to test difference on Flexibility level (Low vs. High). In Table 2 and Table 3, we reported descriptive of neuropsychological and physiological variables, divided in constructs and type measurements used.

Independent *t*-tests were calculated to determine whether two conditions of Flexibility (Low Flexibility, High Flexibility) differed in term of psychophysiological variables. The independent group was divided into two sub-groups a posteriori based on the median of Cognitive Flexibility Inventory (CFI). The use of median depends on the fact that we did not have a normative sample of elderly people. Independent t-tests showed no statistical significance, either for any of the neurophysiological measurements, or for neuropsychological battery, except for scale of Re-Appraisal in ERQ (R ERQ) [t (28) = 2.64, *p* = 0.014, d = 0.97], and almost significant for LF/HF ratio FFT [t (25) = −1.78, *p* = 0.08, d = −0.69].

To collect more information regarding those findings, we conducted computational analyses on physiological and neuropsychological measures, using leave one out cross validation (for additional information about algorithms that were used for the Python computation, please see http://docs.orange.biolab.si/3/data-mining-library/reference/preprocess.html (accessed on 1 July 2023). Dichotomized Cognitive Flexibility Inventory (CFI) self-report was used as predicted variable, divided in Low flexibility and High flexibility [43,44]. Neuropsychological measurements and physiological measurements were used as predictors. The first model took into consideration only neuropsychological measurements, based on neuropsychological measurements of CF and cognitive and affective concurrent variables.

TMT, ACE-AO, ACE-M, ACE-VS, and MOCA are used for neuropsychological measures of Cognitive Flexibility. AES, BDI, HARS, and ERQ assessed affective concurrent variables. CIRS assessed Cognitive Decline, whereas Cognitive Function Instrument, self-report and partner-report scale, measured cognitive functional abilities. The results were not satisfying. It showed a precision between 30% and 60% with most of the loss due to predicted High flex when actual was Low Flex, with an error ranging from 72%, as highlighted in the confusion matrices. ERQ-R is the first predictor in indices ranking (Figure 1).

The second model considered all physiological measures, based on HRV, RSP Rate, f-EMG, and SCR, with the indeces ranked as showed in Figure 2 (http://scikit-learn.org/stable/modules/tree.html (accessed on 1 July 2023)

Hence, we selected the first 10 ranking measurements, testing abilities to discriminate between High flexibility and Low flexibility. The results showed a precision between 64% and 81% (Table 4) with most of the loss due to predicted High Flexibility when actual was Low flexibility, with an error ranging from 20% to 30%, as highlighted in the confusion matrices.

In ranking the 10 measurements, cardiac indices predominated (i.e., LF, HF, VLF, LF/HF ratio, NN50, and Max HR), followed by two measurements of breathing (mean and RMS respiration rate) and one of facial electromyography (RMS EMG2/EMG1). Interestingly high levels of sympathetic activation were found in subjects with low flexibility, as evidenced in linear analysis (Figure 3).

Based on the classification tree and ranking, we tested a third model with only Root Mean Square of Respiration Rate (RMS RSP rate) and Max HR (Figure 4). The results significantly improved and become excellent. The results showed a precision between 60% and 90% (Table 5) with most of the loss due to predicted High Flexibility when actual was Low flexibility, with an error ranging from 10% to 20%, as highlighted in the confusion matrices (Figure 5).

The high discriminative abilities of RMS respiration rate and Max HR are also shown through the Decision tree, a simple algorithm that splits the data into nodes by class purity (information gain for categorical and regression metric for numeric target variable). It is a precursor to the Random Forest algorithm (Figure 6).

Figure 6 shows that those with an RMS RSP Rate greater than 35 in 100% of cases are inflexible. Those with smaller RMS Respiration rate values are 100% flexible if they also have values lower than 93 in Max HR.

## 4. Discussion

Mental flexibility is a complex construct that is still difficult to define today. Generally, it is defined as the subject’s ability to act, react, and think variably. In previous studies, two cognitive and affective components have been outlined and analyzed as part of executive functions, with young adults or patients.

In this study, a healthy aging sample was involved, and both the cognitive and affective components have been jointly measured by integrating conventional neuropsychological tests (usually addressing the cognitive dimension) with continuous psychophysiological signals (used to measure the affective dimension). The aim was to outline the bio-psychological markers of MF especially in healthy aging. To this end, we considered an extensive battery in assessing Cognitive Flexibility: from neuropsychological tests to self-reports, with affective-cognitive measures as measures of convergent validity. Instead, for measures of Affective Flexibility, we considered psychophysiological signals measured during the administration of emotional imagery. This is the first time in which a direct and continuous methodology of assessment of AF—intended as physiological variability—was implemented [4,67].

Our first linear model showed that neither classical neuropsychological tests (TMT, MOCA, or ACE) or physiological measuring (BVP, f-EMG, GSR, or RSP) nor concurrent measures (BDI, AES, CFI, or CIRS) significantly discriminated individual subcomponents between flex vs. low in mental flexibility, except for ERQ Re-appraisal. As a between-grouping variable, we considered CFI scores as a more ecological measure than classical neuropsychological tests, dividing the sample into high and low flexibility. This division allowed us to examine associations with the psychophysiological measures collected.

Results from the neuropsychological tests were consistent with previous studies that showed no significant correlations between neuropsychological measures and self-reports, measuring CF [24,51]. Although formally afferent to the same construct of Cognitive Flexibility, the measures appear to measure different underlying abilities. The CF investigated by neuropsychological tests is related to tasks on executive functions, while Cognitive Flexibility related to self-reports to everyday life situations [18,24,25,68,69]. It is probable that neuropsychological tests provide a direct and controlled measure of cognitive flexibility based on participants’ performance in structured cognitive tasks, but are poorly-generalizable to real-life situations.

Only the Re-appraisal subscale of the Emotion Regulation Questionnaire had significant differences in Low and High Flexibility: subjects with high flexibility had high levels of Re-appraisal abilities. This result is consistent with the literature [13,53,60,70,71,72,73,74]. Emotional regulation, particularly in the form of reappraisal, plays a significant role in mental flexibility. Re-appraisal is a cognitive emotion regulation strategy that involves reinterpreting the meaning of emotional situations to modify one’s emotional response. It allows individuals to adaptively regulate their emotions and flexibly shift their cognitive perspective when confronted with emotionally challenging situations [53,60,73].

Among cardiac, electromyographic, galvanic, and respiratory physiological signals, there is no obvious significance, but some evidence for spectral indices (LF/HF ratio). The LF/HF ratio measures the autonomic nervous system (ANS) activity derived from heart rate variability (HRV) analysis. The ANS has two main branches: the sympathetic nervous system (SNS) and the parasympathetic nervous system (PNS). The LF/HF ratio reflects the balance between sympathetic and parasympathetic activities.

Individuals with lower mental flexibility exhibited higher LF/HF ratio, indicating a higher sympathetic dominance in their autonomic nervous system. The connection between Mental Flexibility and LF/HF ratio suggests that the ability to adapt and shift cognitive strategies is linked to autonomic nervous system regulation. Higher mental flexibility may be associated with a more efficient and adaptive response to stress, promoting better emotional regulation and overall well-being. This finding, also confirmed by nonlinear analyses later, is novel compared with all other studies on Affective Flexibility: no previous study had considered long-term cardiac indices such as spectral ones.

To highlight the significant evidence uncovered by the linear model, we employed multilayer non-linear models, such as Supervised Machine Learning. The rationale behind this decision is that the neuropsychological and physiological components of flexibility may have complex interactions, and these interactions can be better explored through a comprehensive and multivariate approach such as ML, as suggested by Uddin et al. 2021 [10]. Traditional linear models may struggle to capture such complex relationships and might overcall nonlinear associations. It can identify nonlinear and interactive effects, providing a more nuanced understanding of how cognitive and affective processes jointly influence flexibility. This flexibility in model representation is precious in the context of complex human behavior, where linear assumptions may not fully capture the intricacies of cognitive and affective interactions. Additionally, ML offers predictive capabilities, enabling us to develop models that can accurately classify individuals based on their flexibility profiles. This predictive power can be precise in clinical settings, where identifying individuals at risk for cognitive or affective impairments could facilitate early intervention and personalized treatment plans [75].

Hence, the second model proposed a non-linear approach, with supervised Machine Learning (ML), including all neuropsychological measurements (those specific to Cognitive Flexibility and all measures of affective and cognitive convergence). The ML model replicates the considerable ERQ difference: the model included complete neuropsychology testing and ERQ ranked first. Only three of the top 10 ranking indices directly evaluate cognitive flexibility (TMTBA, MOCA, and ACE VS); the rest are concurrent measures of affective states (AES, BDI, and AES), cognitive impairment (CIRS), and global cognitive performance (CFI self). In our hypothesis, these results could depend on the nature of the predicted variable, a self-reported measure of Cognitive Flexibility (CFI). In fact, the literature shows that these self-report questionnaires measure the ability to adapt flexibly by asking the subject to imitate everyday life situations [24,51]. The CFI would represent a more ecological measure of flexibility, managing to integrate affective elements, such as Emotion Regulation, and more cognitive elements, such as Executive Functions. Regardless, the model’s discriminative capabilities remained limited and uninformative.

This, however, cannot be stated for physiological measurements. We implemented a third ML model, which considers all the physiological indices extracted from f-EMG, BVP, RSP, and SCR as predictors. It has discrete discriminative capabilities, and the model clearly improved by picking only the first 10 indices.

Prominent among the findings are primarily cardiac indices, i.e., temporal ones (Max HR and NN50), as indicated by the literature [14], and spectral ones (LF, LF/HF ratio, and VLF), as anticipated by the linear analysis. Overall, the ML model confirms the limited evidence from the linear model: subjects with low mental flexibility appear to have higher sympathetic nervous system activation in response to emotional stimuli, confirmed by significantly higher maximum heart rate (Max HR). They seem more reactive to emotional stimuli, maintaining higher levels of sympathetic activation throughout the experiment, likely linked to emotional dysregulation, as suggested by significantly lower levels of Reappraisal [26,61,76,77,78,79].

In terms of flexibility, RMS values also are novel findings. The root mean square (RMS) is commonly used as a substitute for standard deviation when the input signal has a zero mean, referring to the square root of the mean squared departure of a signal from a baseline or fit. RMS implies a standard deviation or variability in the signal, which fits nicely with the concept of flexibility defined in terms of adaptive variability. This variability was described in terms of electromyographic (RMS EMG2/EMG1) and respiratory rate (RMS RSP). Electromyographic activation is closely related to the valence of the emotional images seen. In particular, the ratio of EMG2/EMG1 expresses greater activation for zygomatic facial motion than for corrugator facial motion. Greater variability in activation of the two facial movements could aid in discriminating between High and Low Flexibility. Respiration rate, expressed both as mean and RMS, is one of the most important indices in the ranking.

The last Machine Learning (ML) model only considers the top two of ten ranking indicators (Max HR and RMS RSP), and it has the highest discriminatory ability between high and low flexibility. Max HR [beats/min] and RMS respiration rate shed light on how physiological parameters could interact and discriminate between High and Low Flexibility: the variability of those with lower levels of flexibility is connected to the presence of polarization at the upper and lower extremes, as the decision tree and scatterplot show (Figure 5 and Figure 6). On the other hand, those with high flexibility have scores that are distributed evenly on average. Those with low cognitive flexibility have extreme values in the high and low ranges, deceptively creating greater data dispersion. This may help to explain earlier research results showing higher cardiac variability in people with lower cognitive flexibility. As the scores of persons with low Flexibility are bipolar at the extremes, short-term markers such as Max HR and RMSSD will likely show variability. On the other hand, spectral indices are less impacted because they steal more time.

In conclusion, our findings suggest that physiological measures demonstrate superior discriminative capabilities between high and low flexibility compared to neuropsychological tests, highlighting the importance of considering comprehensive bio-physiological markers to understand better the complexities of flexibility in the context of cognitive and affective processes.

One possible explanation for the superior discriminative capabilities of physiological measures is their ability to capture real-time responses to emotional and cognitive stimuli. Physiological indices, such as heart rate variability and electromyographic activation, directly reflect the autonomic nervous system’s dynamic changes in response to emotional experiences and cognitive demands. In contrast, traditional neuropsychological tests primarily assess cognitive abilities in controlled settings, which may only partially capture the complex and adaptive nature of flexibility in real-life scenarios.

Moreover, physiological measures offer a more holistic representation of an individual’s emotional and cognitive reactivity, as they integrate both cognitive and affective aspects of flexibility. For instance, heart rate variability can reflect the interplay between cognitive appraisal and emotional regulation during adaptive responses to challenges. This comprehensive approach allows physiological measures to capture the interwoven complexities of flexibility, providing a more accurate and nuanced depiction of an individual’s flexibility profile.

Furthermore, physiological measures offer the advantage of being less susceptible to conscious control or cognitive biases that can influence self-report measures used in traditional neuropsychological assessments. As a result, physiological indices may provide a more objective and reliable assessment of an individual’s cognitive-affective responses, enhancing their capacity to distinguish between high and low flexibility.

Despite the enhanced discriminative capabilities of physiological measures, it remains crucial to consider and integrate both physiological and neuropsychological assessments in the comprehensive evaluation of flexibility, as they collectively provide a more comprehensive and multidimensional understanding of the cognitive-affective aspects underlying this complex construct.

We hypothesize that considering the numerous behavioral and physiological markers in a complex non-linear model has allowed relationships that otherwise would not have occurred.

## 5. Conclusions

Mental flexibility, with its cognitive-affective and physiological biomarkers, is a complex and multifaceted construct. Studying its components in healthy aging enables a better understanding of how they work and interact, as a signal of successful aging [59].

Our findings highlight how just two physiological parameters, cardiac variability (Max HR) and RMS Respiration Rate, can discriminate between high and low flexibility. In general, physiological measurements are more discriminating and predictive than the neuropsychological test batteries usually used. Affective flexibility, on the other hand, measured continuously through physiological measures taken during stimulus display, is a sound methodology of analysis that captures the full range of changes in behavioral-physiological typical of flexibility.

This study shows that nonlinear multivariate models can disclose the complicated relationship between cognitive and emotional flexibility and physiological data. Future studies could use Virtual Reality (VR) or 360° videos to test cognitive and affective flexibility jointly [80]. Both could allow ecological flexibility studies by placing participants in real-life scenarios where they must make behavioral, relational, and decision-making judgments. VR or 360° environments can elicit cognitive and affective responses, providing valuable insights into how individuals adapt and shift cognitive strategies in dynamic and complex ecological settings, which are similar to the equivalent real ones. Furthermore, they enable controlled manipulation of stimuli and scenarios, facilitating systematic investigations of flexibility under various conditions. One potential concern about VR is the cybersickness or discomfort experienced by participants, which may impact cognitive performance and emotional responses. Instead, for 360° videos the quality and realism of the videos vary, potentially impacting participant engagement and response validity. Using either tool depends on the choice of scenarios, whether the participant can interact or only observe, budget considerations, and the desired level of realism to be achieved.

Finally, both would enable us to investigate flexibility in its two components continuously integrate rather than splitting them into mechanisms or sub-mechanisms.

## 6. Limitations

The study investigated biopsychological markers of cognitive-affective flexibility in healthy aging using a sample of 35 participants aged 62–82 years, with a gender imbalance, as approximately 77% were female. However, this gender unbalance reflects the national demographic trend of healthy ageing population, showing a higher number of females than males (the percentage of female vs. male is 63% to 37%). One potential limitation could be gender distribution, which may impact the generalizability of the findings. A more balanced sample should be considered in further studies. Another limitation was the absence of a control group, such as socially inactive individuals or those with neurological conditions, to compare flexibility biomarkers between different populations. Therefore, future studies could consider this group comparison. However, our study minimizes this lack, with more than 100 variables per subject, including neuropsychological tests and physiological measures, ensuring high representativeness of intrasubject and between-subject variability. Although the cross-sectional design provided a snapshot of flexibility at a specific time, a longitudinal approach could be beneficial for understanding its developmental trajectory. Therefore, a further step could take into consideration longitudinal analysis on the same healthy elderly group, to trace flexibility changes (neuropsychological and physiological) over the time.

Additionally, the study adopted COVID-related images as emotional stimuli, which are very specific to the period in which the study was conducted; thus, this study could be replicated with images reporting a broader range of emotional contents. Despite these limitations, the study offered key and novel insights into cognitive-affective flexibility in healthy aging and suggested directions for future investigations.

## Figures and Tables

**Figure 1 sensors-23-06983-f001:**
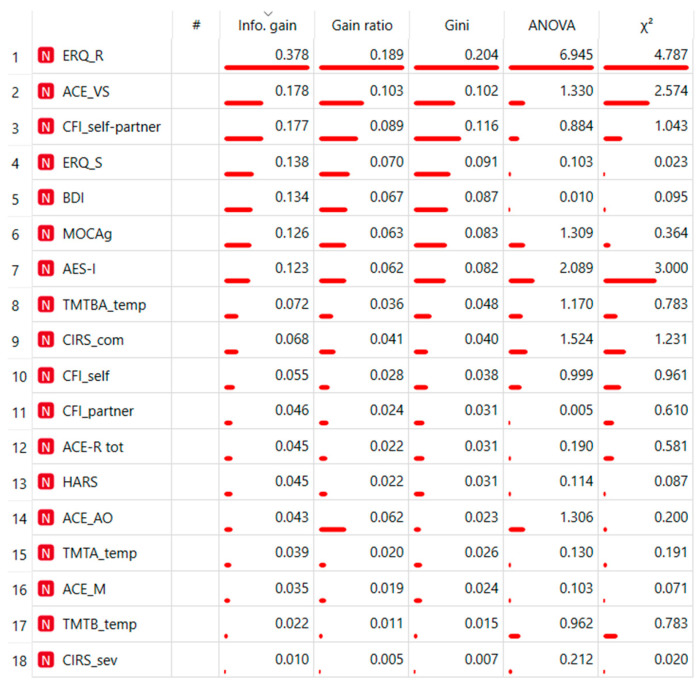
Ranking of Neuropsychological measurement discriminating low vs. high mental flexibility.

**Figure 2 sensors-23-06983-f002:**
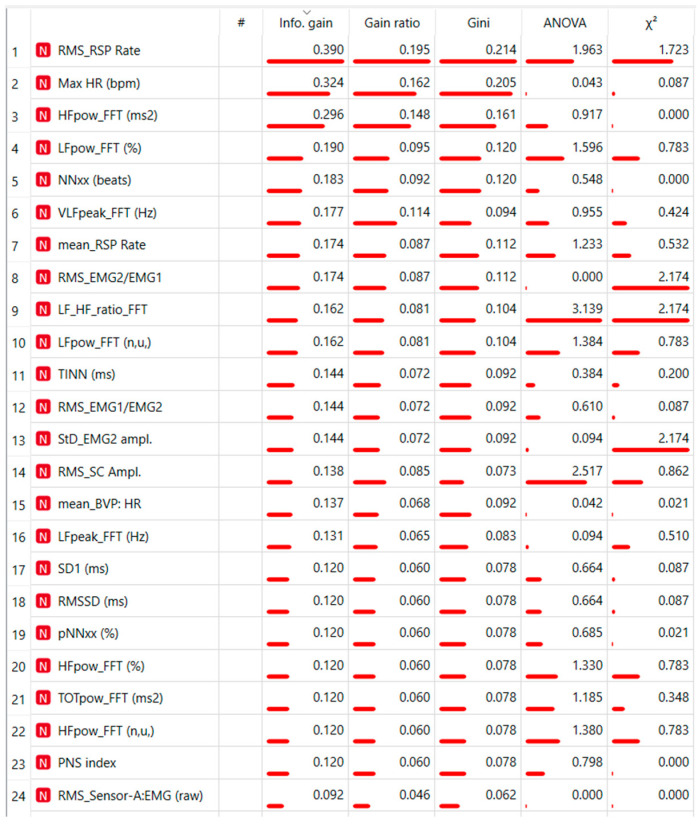
Indexes Ranking of all neurophysiological measurements.

**Figure 3 sensors-23-06983-f003:**
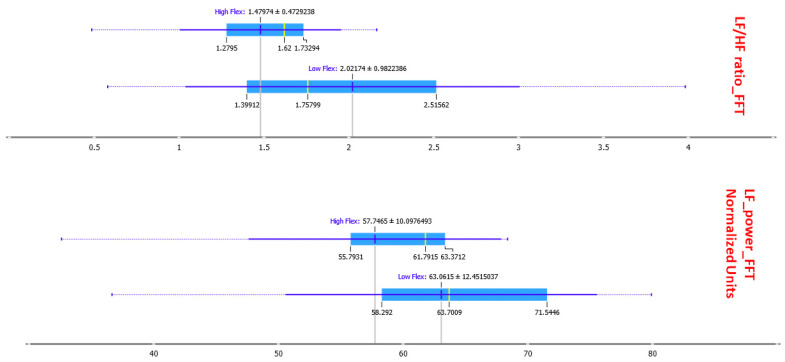
LF/HF ratio and LF power between High Flexibility and Low Flexibility.

**Figure 4 sensors-23-06983-f004:**
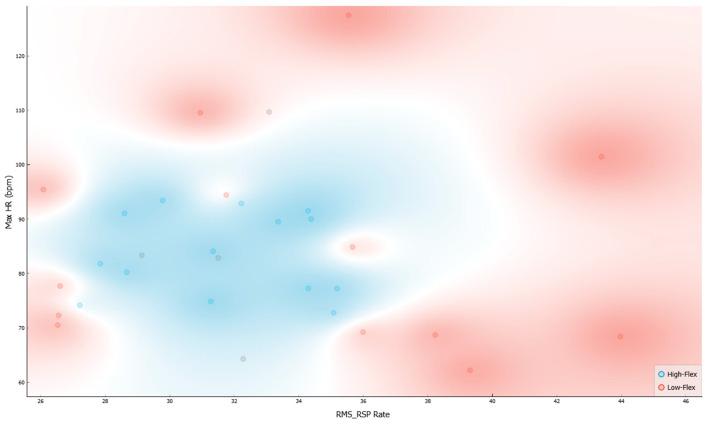
Scatterplot between High and Low Flexibility for Max HR and RMS respiration rate (RMS RSP). External part shows low flexibility level vs Internal part shows high flexibility level.

**Figure 5 sensors-23-06983-f005:**
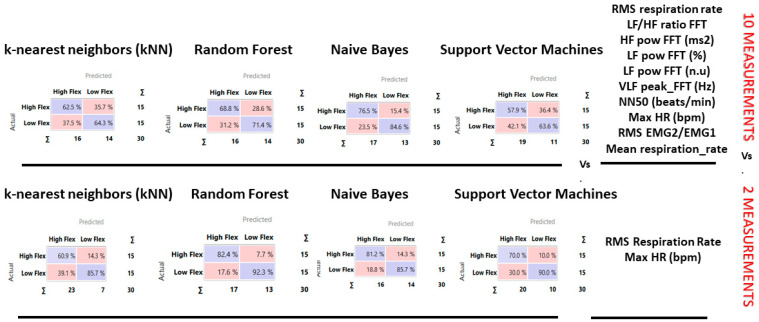
Confusion Matrix for the four classification methods of two models: 10 vs. 2 physiological measurements. The diagonal values (i.e., purple boxes) represent the correct proportion predicted, while of-diagonal elements are those that are mislabeled by the classifier.

**Figure 6 sensors-23-06983-f006:**
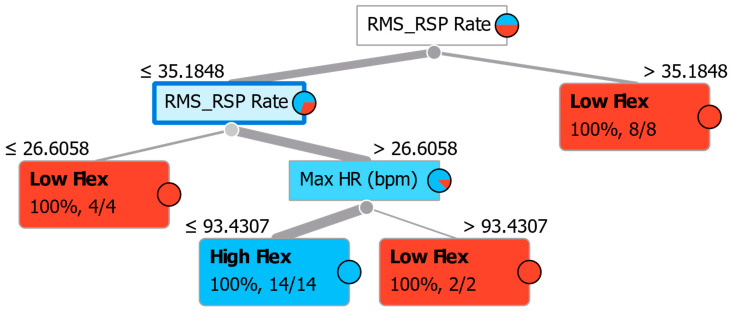
Decision tree for discrimination between Low and High Flexibility. Small circles indicate the ratio of classifications reported inside the rectangle in terms of percentage of correctness in recognizing the specific characteristics. The colors indicate classification as one of the two groups: blue for High flexibility subjects, red for Low flexibility subjects.

**Table 1 sensors-23-06983-t001:** Neuropsychological battery.

**Flexibility Measures**	**Construct**	**Neuropsychological Measurements**	**Acronym**
	Cognitive Flexibility	Trail Making Task	TMT A TMT B TMT BA
	Cognitive Flexibility	Addenbrooke’s Cognitive Examination Revised version: Memory, Visual spatial, Attention/Orientation scale	ACE AO, ACE M, ACE VS
	Cognitive Flexibility	Montreal Cognitive Assessment	MOCA
	Cognitive Flexibility	Cognitive Flexibility Inventory	CFI
**Convergence Measures**	Decline cognitive	Cumulative Illness Rating Scale: Comorbidity and Severity scale	CIRS COM, CIRS SEV
	Functional Abilities	Cognitive Function Instrument: self-report, partner, self-partner	CFI self, CFI partner, CFI self-partner
	Depression	Beck Depression Inventory	BDI
	Apathy	Apathy Evaluation Scale	AES
	Anxiety	Hamilton Anxiety Rating Scale	HARS
	Emotion Regulation	Emotion regulation Questionnaire: Re-appraisal Suppression	ERQ R ERQ S

**Table 2 sensors-23-06983-t002:** Neuropsychological measurements.

Constructs	Type	Test	Group	N	Mean	SE	SD
**Cognitive flexibility**	**Neuropsychological test**	TMTA	High Flex	15	38	3.2	12.4
			Low Flex	15	37.2	2.54	9.84
		TMTB	High Flex	15	100.47	12.9	50
			Low Flex	15	89.8	5.11	19.8
		TMTBA	High Flex	15	62.6	10.5	40.5
			Low Flex	15	52.6	4.27	16.5
		ACE-R	High Flex	15	94.73	0.54	2.09
			Low Flex	15	94.6	0.7	2.72
		ACE AO	High Flex	15	17.67	0.21	0.82
			Low Flex	15	17.93	0.07	0.26
		ACE M	High Flex	15	24.8	0.26	1.01
			Low Flex	15	24.67	0.39	1.5
		ACE VS	High Flex	15	15.27	0.27	1.03
			Low Flex	15	14.8	0.22	0.86
		MOCA	High Flex	15	26.4	0.68	2.64
			Low Flex	15	27	0.56	2.17
**Cognitive Function**	**Self-report**	CFI self	High Flex	15	2.57	0.45	1.74
			Low Flex	15	3.2	0.45	1.73
		CFI partner	High Flex	15	1.37	0.26	1.03
			Low Flex	15	1.33	0.37	1.44
		CFI self-partner	High Flex	15	1.2	0.53	2.04
			Low Flex	15	1.87	0.47	1.84
**Cognitive decline**	**Self-report**	CIRS sev	High Flex	15	1.33	0.04	0.17
			Low Flex	15	1.36	0.03	0.14
		CIRS com	High Flex	15	1.13	0.19	0.74
			Low Flex	15	1.47	0.22	0.83
**Affective**	**Self-report**	BDI	High Flex	15	6.33	1.63	6.31
			Low Flex	15	6.87	1.67	6.48
		HARS	High Flex	15	7.6	1.37	5.3
			Low Flex	15	7.33	1.41	5.46
		ERQ R	High Flex	15	32.07	1.36	5.28
			Low Flex	15	27.4	1.13	4.37
		ERQ S	High Flex	15	17.47	1.04	4.02
			Low Flex	15	18	1.29	5.01

**Table 3 sensors-23-06983-t003:** Physiological Measurements.

Physio	Type	Measurements	Group	N	Mean	SE	SD
**Heart Rate Variability**	**Temporal domain**	RMSSD (ms)	High Flex	14	25.12	2.18	8.14
			Low Flex	13	37.70	15.91	57.37
		SDNN (ms)	High Flex	14	24.01	2.18	8.16
			Low Flex	13	33.04	10.10	36.43
		Max HR (bpm)	High Flex	14	83.62	2.04	7.62
			Low Flex	13	84.78	5.44	19.60
		NN50 (beats)	High Flex	14	178.50	44.95	168.20
			Low Flex	13	293.00	153.65	553.98
	**Frequency domain**	VLF pow FFT (ms^2^)	High Flex	14	43.99	12.00	44.91
			Low Flex	13	70.89	24.12	86.98
		LF pow FFT (ms^2^)	High Flex	14	305.53	66.88	250.22
			Low Flex	13	934.41	529.39	1908.74
		HF pow FFT (ms^2^)	High Flex	14	222.51	59.00	220.76
			Low Flex	13	806.51	631.68	2277.57
		LF HF ratio FFT	High Flex	14	1.48	0.13	0.49
			Low Flex	13	2.02	0.28	1.02
	**Non linear**	SD1 (ms)	High Flex	14	17.76	1.54	5.76
			Low Flex	13	26.66	11.25	40.58
		SD2 (ms)	High Flex	14	28.76	2.83	10.58
			Low Flex	13	37.40	9.16	33.02
		SD2 SD1 ratio	High Flex	14	1.65	0.10	0.36
			Low Flex	13	1.79	0.13	0.46
**Facial Elettromiography**		mean EMG1/EMG2	High Flex	14	2.99	0.38	1.44
			Low Flex	14	3.15	0.39	1.46
		mean EMG2/EMG1	High Flex	14	1.58	0.90	3.39
			Low Flex	14	1.97	1.17	4.38
		StD EMG1/EMG2	High Flex	14	3.83	2.25	8.43
			Low Flex	14	9.07	6.17	23.08
		StD EMG2/EMG1	High Flex	14	19.81	14.76	55.23
			Low Flex	14	19.75	17.65	66.03
		RMS EMG1/EMG2	High Flex	14	5.50	2.17	8.12
			Low Flex	14	10.52	6.06	22.69
		RMS EMG2/EMG1	High Flex	14	20.08	14.77	55.25
			Low Flex	14	20.05	17.67	66.11
**Skin Condactance**		Mean SC	High Flex	13	2.38	1.37	0.38
			Low Flex	14	2.35	0.92	0.25
		StD SC	High Flex	13	0.30	0.23	0.06
			Low Flex	14	0.29	0.13	0.04
		RMS SC	High Flex	13	2.41	1.37	0.38
			Low Flex	14	2.37	0.92	0.25
**Respiration Rate**		Mean RSP Rate	High Flex	14	29.83	0.80	3.00
			Low Flex	14	31.85	1.64	6.14
		StD RSP Rate	High Flex	14	10.52	0.44	1.66
			Low Flex	14	12.26	0.87	3.27
		RMS RSP Rate	High Flex	14	31.68	0.76	2.84
			Low Flex	14	34.26	1.67	6.26

**Table 4 sensors-23-06983-t004:** A leave one out cross-validation: Four learning algorithms were compared, i.e., (1) kNN, (2) Support vector machine, (3) Random Forest, and (4) Naïve Bayes.

Method	AUC	CA	F1	Precision	Recall
**kNN**	0.54	0.63	0.63	0.63	0.63
**Support Vector Machine (SVM)**	0.46	0.60	0.59	0.61	0.60
**Random Forest**	0.68	0.60	0.60	0.60	0.60
**Naive Bayes**	0.81	0.80	0.80	0.81	0.80

**AUC** (Area under the ROC curve) is the area under the classic receiver-operating curve. **CA** (Classification accuracy) represents the proportion of the classified examples correctly. **F1** represents the weighted harmonic average of the precision and recall (defined below). **Precision** represents a proportion of true positives among all the instances classified as positive. In our case, the proportion of a condition was identified correctly. **Recall** represents the proportion of true positives among the positive instances in our data.

**Table 5 sensors-23-06983-t005:** A leave one out cross-validation: Four learning algorithms were compared, i.e., (1) kNN, (2) Support vector machine, (3) Random Forest, and (4) Naïve Bayes.

Method	AUC	CA	F1	Precision	Recall
**kNN**	0.68	0.67	0.64	0.73	0.67
**Support Vector Machine (SVM)**	0.77	0.77	0.76	0.80	0.77
**Random Forest**	0.89	0.87	0.87	0.87	0.87
**Naive Bayes**	0.88	0.83	0.83	0.83	0.83

## Data Availability

The datasets used and/or analysed during the current study is available from the corresponding author on reasonable request.

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
