# Peer review of "Exploring Biomarkers of Mental Flexibility in Healthy Aging: A Computational Psychometric Study"

_sensors, 2023, doi:10.3390/s23156983_

Round 1

Reviewer 1 Report

Thank you for the opportunity to revise that paper. The manuscript needs some work so that I can recommend its publication.

Major Points:

Abstract

Overall, the abstract effectively presents the rationale, objectives, and potential contributions of the study. It also demonstrates an understanding of the existing literature and highlights the novelty of the study by focusing on healthy older subjects. One improvement that could be made is to provide a brief explanation of what is meant by "non-linear multi-variate analytic approaches." This would help clarify the methodological approach used in the study.

Introduction

1.      Overall, I think the introduction was done hastily and lacks conceptual structure.

2.      However, the authors provide a clear and concise overview of the concept of Mental Flexibility (MF) and its components, Cognitive Flexibility (CF) and Affective Flexibility (AF). It highlights the importance of these components in adjusting goals and shifting thoughts or actions according to changing demands of the situation. Additionally, it presents the different ways in which CF and AF are measured and discusses the limitations of previous studies in terms of their focus on executive functions, incomplete assessment of AF, and sampling of specific populations.

3.      There are some areas in which the introduction could be improved. Firstly, the introduction lacks a clear research question or objective. It would be beneficial to explicitly state what the study aims to achieve, such as investigating the relationship between cognitive-affective flexibility and healthy aging or identifying psycho-bio markers of flexibility in older adults. This would provide the reader with a clearer understanding of the purpose of the study. Secondly, there is repetition of certain information, particularly regarding the lack of significant correlation between CF measures and CF self-reports. While it is important to highlight this discrepancy, it could be condensed to avoid redundancy in the introduction.

Lastly, the introduction could benefit from more contextualization of the topic. For instance, it would be helpful to explain why understanding cognitive-affective flexibility in healthy aging is important and how it can impact individuals' well-being or quality of life. Providing this context would create a stronger rationale for conducting the study.

In sum, the introduction could be strengthened with a clearer research question or objective, condensing repetitive information, and providing more contextualization. There are no predictions from the study.

Methods:

1.      Sample size: The study only included 35 participants, which might be considered a very small sample size.

2.      Participant characteristics: While the age range of the participants (62-82 years) aligns with the target population of older adults, the sample consisted entirely of socially active individuals from the University of the Third Age. This might limit the generalizability of the findings to a broader population of older adults who may have different levels of social engagement and activities.

3.      Exclusion criteria: While the study states that participants were not taking any psychotropic medications that could have affected their cognitive abilities or mood, it does not provide information on how this was assessed or verified. Lack of objective measures or screening tools to confirm this information could be a limitation.

4.       Inclusion criteria for frailty: The study mentions that none of the participants met Fried et al.'s inclusion criteria for frailty status, but it does not provide further details on what these criteria are. Without this information, it is difficult to evaluate the potential impact of frailty on the study results.

5.      Procedure: The description of the procedure lacks specific details about the recruitment process, how participants were selected, and how they were contacted. This lack of information raises potential concerns about the representativeness of the sample and the potential for selection bias.

6.      Neuropsychological battery: The study uses various measures to assess cognitive flexibility, but the rationale for selecting these specific measures is not clearly explained. Additionally, it does not provide information on the reliability and validity of these measures or whether they have been previously validated in a similar population.

7.      Psychophysiological test: The description of the psychophysiological test to assess affective flexibility is limited. It does not provide details on the stimuli used, the specific measurements obtained, or the protocols followed during the testing. Without this information, it is difficult to evaluate the scientific rigor of the test.

8.       Lack of control group: The study does not mention the inclusion of a control group. Having a control group would have allowed for a comparison between the socially active older adults and a group of older adults who are not socially active. This comparison would have provided valuable insights into the potential effects of social activity on cognitive and affective flexibility.
In summary, while the described method provides a general overview of the study design, it lacks some important details and may have potential limitations in terms of sample size, participant characteristics, inclusion criteria, and the lack of a control group. These limitations should be considered when interpreting the study findings and generalizing them to the broader population of older adults.

Results

1.      They are clear and appear conveniently described.

Discussion:

It doesn't make much sense to me to occupy the discussion section with the results.

Overall, the discussion presents some interesting conclusions regarding mental flexibility and its components in healthy aging. However, there are a few potential limitations and areas for improvement that should be considered.

Firstly, while the study suggests that physiological measurements are more effective at discriminating between high and low flexibility than traditional neuropsychological tests, it does not provide a clear rationale for why this might be the case.

Further explanation or discussion on why physiological measurements are more accurate or reliable would strengthen the overall argument.

Additionally, the analysis mentions the continuous measurement of affective flexibility through physiological measures during stimulus display as a sound methodology. While this may be a valid approach, it would be helpful to include some examples or specific studies that have used this methodology successfully.

This would provide further evidence for the reliability and validity of this approach.
Furthermore, the analysis suggests that nonlinear multivariate models can reveal the relationship between cognitive and emotional flexibility and physiological data. While this is an intriguing idea, it would be beneficial to provide some specific examples or studies that have utilized these models effectively. This would give readers a better understanding of how this approach can be successfully applied in practice.

Lastly, the analysis proposes that future studies could use Virtual Reality to test cognitive and affective flexibility in real-life scenarios. While this is a promising suggestion, it would be helpful to discuss potential challenges or limitations that may arise when using Virtual Reality. This would provide a more well-rounded perspective and ensure that researchers are aware of any potential issues they may encounter. In conclusion, this analysis provides some interesting insights into mental flexibility and its components in healthy aging. However, further explanations, examples, and considerations would strengthen the overall argument and provide a more comprehensive understanding of the topic.

Author Response

Dear Reviewer, please find enclosed the responses to your comments.

Reviewer 2 Report

The authors present a study exploring biomarkers of mental flexibility in healthy aging. The manuscript consists of 19 pages, including 5 tables, 6 figures and 58 references.

Comments and suggestions:

Materials and Methods:

-          35 people were included in the study. Age ranges from 62-82 years and 27 participants were female (~77%). Study population is very small and there is a clear gender imbalance.

-          Did the authors perform a sample size calculation, which justifies that low number of participants?

-          Authors state that “socially active people from University of the Third Age (UNITRE) in Turin (Italy)” participated. What relationship do these people have with this institution? Are they patients? With the given information, the study population is not sufficiently described, it is essential to formulate clear inclusion and exclusion criteria.

-          Are there any reasons for the greater number of female participants?

-          In lines 124-125 abbreviations (ACE-AO, ACE-M, ACE-VS, MOCA) are not explained.

Results and Discussion:

-          Presentation of the results should be revised. With the current presentation, I have difficulties to detect the significant findings. In a study containing so much information, I miss a common thread and a high lightening of the main results.

-          What are the reasons to include figures (scatterplot, decision tree and LF/HF ratio) in the discussion section? This is very confusing, for usual that information belongs to results section.

-          The manuscript should be adapted with information about strengths and particularly with limitations of the presented study. This information is completely missing.

Author Response

(The authors gave the same response as above.)

Round 2

Reviewer 1 Report

The authors have made a set of improvements to the document. I leave it to the editor's discretion to accept the manuscript, evaluating the relevance and robustness of the results.

Author Response

Dear Reviewer,

Thank you very much for your time and dedication in reviewing our paper. We truly appreciate your effort and insightful feedback. Thanks to your valuable suggestions, we have made significant improvements to the manuscript.

Reviewer 2 Report

I absolutely appreciate authors’ efforts improving the manuscript. 

I have one last suggestion:

-       In authors' responses to my comments sample size calculation is described. I suggest manuscript adaptation with this information to make the small sample size more comprehensible for the interested reader.

Author Response

Thank you for your valuable feedback on our manuscript. We appreciate your careful review and your suggestion to make the small sample size more comprehensible to the interested reader.

To follow up on your comment, we have seriously considered your suggestion. We will now include a more detailed description of the sample size calculation in the manuscript. This addition will give the interested reader a clearer understanding of how the sample size was determined and the reason why it is. Once again, we thank you for your thoughtful input, which has undoubtedly contributed to the overall quality of our paper.

We added this part in 111-118 lines:" A group of 35 (27 women; 62–82 years) socially active older people from the University of the Third Age (UNITRE) in Turin (Italy) were recruited and all agreed to participate. A priori power analysis was conducted using G∗Power version 3.1.9.7 (35) to determine the minimum sample size required to test the study hypothesis, based on Fernandez-Aguilar et al., (36). Results indicated the required sample size to achieve 95% power for detecting a large effect, at a significance criterion of α = 0.05, was N = 26 for independent samples t-test (High Flex vs. Low Flex). Considering dropouts and recording issues with physiological signals we recruited 35 participants".